# Recent Progress in Second Near-Infrared (NIR-II) Fluorescence Imaging in Cancer

**DOI:** 10.3390/biom12081044

**Published:** 2022-07-28

**Authors:** Tian Wang, Yingying Chen, Bo Wang, Xiaofan Gao, Mingfu Wu

**Affiliations:** Department of Obstetrics and Gynecology, Tongji Hospital, Tongji Medical College, Huazhong University of Science and Technology, Wuhan 430074, China; wangtian9@tjh.tjmu.edu.cn (T.W.); chenyy@hust.edu.cn (Y.C.); d201981680@hust.edu.cn (B.W.); gaoxf@hust.edu.cn (X.G.)

**Keywords:** cancer, second near-infrared (NIR-II) fluorescence imaging, fluorescent probe, cancer theranostics, cancer diagnosis, cancer treatment

## Abstract

Cancer continues to be one of the leading causes of death worldwide, and its incidence is on the rise. Although cancer diagnosis and therapy have advanced significantly in recent decades, it is still a challenge to achieve the accurate identification and localization of cancer and to complete tumor elimination with a maximum preservation of normal tissue. Recently, second near-infrared region (NIR-II, 1000–1700 nm) fluorescence has shown great application potential in cancer theranostics due to its inherent advantages, such as great penetration capacity, minimal tissue absorption and scattering, and low autofluorescence. With the development of fluorescence imaging systems and fluorescent probes, tumor detection, margin definition, and individualized therapy can be achieved quickly, enabling an increasingly accurate diagnosis and treatment of cancer. Herein, this review introduces the role of NIR-II fluorescence imaging in cancer diagnosis and summarizes the representative applications of NIR-II image-guided treatment in cancer therapy. Ultimately, we discuss the present challenges and future perspectives on fluorescence imaging in the field of cancer theranostics and put forward our opinions on how to improve the accuracy and efficiency of cancer diagnosis and therapeutics.

## 1. Introduction

Cancer is one of the most fatal diseases and leading causes of human death around the world, and its prevention remains a serious issue [1]. Due to the inefficient sensitivity, specificity, and resolution of traditional diagnostic modalities, early-stage tumors, especially tiny lesions, cannot be effectively detected [2,3,4]. Therefore, most patients are diagnosed with cancer at the middle or advanced stage, missing the best time for treatment. Meanwhile, the current clinical treatment strategies for cancer, such as surgery, chemotherapy, and radiotherapy, cause iatrogenic damage and have side effects on the human body, limiting, to a certain extent, the accuracy and efficiency of cancer therapy. Although the treatment methods have advanced in recent decades, there is no significant improvement in survival [5,6,7,8]. Thus, it is urgent to exploit novel cancer theranostics in order to improve the survival and cure rates for cancer patients.

Imaging technology plays a crucial role in the field of cancer theranostics [9,10]. Although traditional imaging techniques, such as ultrasound imaging (US), computed tomography (CT), magnetic resonance imaging (MRI), positron emission tomography (PET), and others have emerged, great challenges still exist for rapid, real-time, high resolution, noninvasive and economic cancer imaging in the complex human body [11]. Fluorescence imaging has aroused considerable interest in the biomedical and life science fields due to its high sensitivity and resolution as well as the simplicity of operation [12]. With the development of optical imaging techniques, the bioimaging spectrum has gradually extended from the visible region (400–700 nm) to the first near infrared (700–900 nm) region and the second near-infrared (1000–1700 nm) region. NIR light has lower tissue absorption, scattering, and autofluorescence, allowing for a deeper penetration depth than visible light [13]. To date, the NIR-I fluorescent probe, indocyanine green (ICG), has been approved by the US Food and Drug Administration (FDA) for in vivo application and clinical diagnosis. The commercially available ICG probe has been widely used for lymph nodes tracking in breast and gastric cancers [14,15]. However, the increasing demand for high precision imaging puts forward higher requirements for fluorescence imaging with a stronger penetration capacity and higher resolution.

Compared with NIR-I fluorescence imaging, NIR-II fluorescence imaging has longer excitation and emission wavelengths, thus exhibiting superior performance in terms of the imaging depth and spatio-temporal resolution [16]. NIR-II fluorescence imaging is expected to provide great help for precise cancer identification and location. Additionally, cancer diagnosis can be combined with phototherapy to achieve the integration of cancer diagnosis and treatment, improving the efficiency and accuracy of cancer therapy [17,18]. In this review, we attempt to summarize the recent advances of NIR-II fluorescence imaging in cancer diagnosis and theranostics (Figure 1)and discuss the current challenges as well as perspectives for the near future.

## 2. NIR-II Fluorescent Probes

Fluorescent probes are fluorescent molecules that fluorescence properties (such as excitation and emission wavelength, intensity, lifetime, and polarization) can change flexibly according to the environment; this can benefit qualitative and quantitative analysis. NIR-II fluorescent probes are considered to be the most promising tools for accurate diagnosis and efficient treatment in cancer, due to the advantages of deep tissue penetration, high spatio-temporal resolution, and low destructive effects [19]. NIR-II fluorophores include two types: inorganic and organic materials. According to the existing literature, each type of the fluorophores has its own advantages and disadvantages (Table 1). Organic materials include organic small molecules and conjugated polymers. Inorganic nanomaterials include quantum dots (QDs) [20], carbon nanotubes (CNTs) [21], and rare earth doped nanoparticles (RENPs) [22]. Inorganic NIR-II fluorescent probes have narrow bandgap energy levels and a high fluorescence quantum yield (QY), but they are difficult to excrete in vivo and have potential long-term biological toxicity. Compared with inorganic materials, organic small molecules and conjugated polymers have great advantages in terms of biocompatibility and biosafety. More importantly, their photophysical properties, such as fluorescence, can be controlled by rational chemical structure design. The organic small molecule NIR-II fluorescent probes reported in the existing literature include cyanine, fluoroborondipyrrole dyes (BODIPY), and the conjugated small molecules of the donor-acceptor-donor (D-A-D) structure [23,24,25]. Organic small molecule probes have excellent properties, such as a rapid metabolism, low toxicity, and clear structure. Their development and application may promote the early diagnosis and precise treatment of tumors and other diseases [26]. However, their production is complex, often requiring multiple synthesis steps and complex separation processes. Organic conjugated polymers have the advantages of a simple preparation, high photochemical stability and extinction coefficient, and strong flexibility of absorption and fluorescence ranges (through the selection of D and A monomers or the regulation of polymerization degree) and have become important candidates for use in NIR-II fluorescence imaging probes [27,28,29].

## 3. NIR-II Fluorescence and Cancer Diagnosis

### 3.1. Tumor Detection

Fluorescence imaging is a powerful tool that enables researchers and physicians to observe tissue directly. The in vivo fluorescence detection of tumors is currently available using nonspecific and specific tumor targets. At present, a large number of tumor markers have been proved to be ideal target indicators for tumor diagnosis and treatment, such as the folate receptor (FR) [30], integrin receptor [31], epithelial growth factor receptor (EGFR) [32], transferrin receptor [33], human epithelial growth factor receptor (HER2) [34], transporter, endothelin receptor [35], angiogenesis, prostate-specific membrane antigen (PSMA) [36], somatostatin receptor [37], gastrin-releasing peptide receptor (GRPR) [38], and peptides [39].

In 2016, Alexander L Antaris et al. synthesized the first NIR-II organic small molecule (CH1055) based on the D-A-D structure, which had excellent water solubility and could be quickly excreted through the kidneys [40]. Moreover, CH1055 showed a strong fluorescence emission peak at 1055 nm. Then, CH1055 was closely conjugated to an anti- EGFR, enabling tumor-targeted molecular imaging in vivo. The T/NT ratio of this probe for the NIR-II fluorescence imaging of human squamous cell carcinoma (SCC) tumors is as high as five, which is higher than the tumor imaging quality of the passive enhanced permeability and retention (EPR) effect [41]. In 2017, Dan Yang et al. [42] synthesized the D-A-D fluorescent dye FEB for the first time, which exhibited a high fluorescence quantum yield and could be successfully applied to tumor imaging in vivo. Besides the D-A-D structure, the fluorescent material FEB also contains a pair of twisted groups and four long alkyl chains, reducing the intermolecular packing and improving the fluorescence quantum efficiency. There are several advantages to FEB dyes, including a large Stokes shift (120 nm), low toxicity, and high stability. Furthermore, FEB dyes are easily modified with PEG or PEG-iRGD to yield FEB-2000 or FEB-2000-iRGD. The FEB-2000-iRGD presents enhanced tumor targeting abilities for the purpose of specifically binding FEB to the tumor, resulting in better tumor imaging quality, indicating that FEB has become a promising tumor imaging agent.

However, it is a major problem of D-A-D organic small molecule fluorescent probes in NIR-II molecular imaging that the fluorescence quantum efficiency is lower because of the significant non-radiative energy transition and fluorescence quenching effect when small organic molecules interact with water molecules in a physiological environment, and this limits the wide application of NIR-II organic small molecule probes [43]. Therefore, it is extremely crucial to enhance the fluorescence quantum yield of small organic molecules and to further increase the T/NT ratio of tumors for early tumor detection and therapy. In order to solve these problems, researchers have further introduced a new shielding unit (S) into the D-A-D structure to form a novel structure of shielding unit-donor-acceptor-donor-shielding unit (S-D-A-D-S), then twisted the conjugated structure to reduce intermolecular interactions, obtaining NIR-II organic small molecule fluorescent probes with a high quantum yield. Rui Tian et al. [44] developed a NIR-II organic fluorescent probe IR-BEMC6P that was synthesized based on the S-D-A-D-S structure; it was a new type of NIR-II molecular fluorescent probe with high fluorescence quantum efficiency and low hepatic uptake, and it could be rapidly excreted by the kidneys. When the IR-BEMC6P@RGD peptide conjugates were injected intravenously, the tumor was able to be distinguished at a T/NT ratio of around six using NIR-II whole body imaging.

Guosong Hong et al. [45] developed a new type of conjugated polymer fluorophore with an absorption range over 1000 nm. The synthetic compound 5H5 showed an absorption peak at 1069 nm under NIR-II excitation and an emission wavelength at 1125 nm. The NIR-II/NIR-IIa imaging at 808 nm/1064 nm excitation was able to be directly compared by absorption and emission spectroscopy. The NIR-IIa vessel and tumor imaging with 1064 nm laser excitation revealed a higher resolution and signal-to-noise ratio compared to imaging near 1000 nm under 808 nm excitation, revealing the potential for the application of small molecule NIR-II fluorophores in in vivo bioimaging. Sun Y et al. [46] combined the NIR-II fluorescent probe H1 with the RGD peptide and explored targeted tumor imaging in U87MG tumor-bearing mice. The ex vivo NIR-II signal indicated that the uptake of H1@RGD in tumors was higher than that in other normal tissues, except in the liver/kidney. Weizhi Wang et al. [47] developed a CD133-binding NIR-II probe for tumor imaging in vivo, and screened a novel CD133 peptide (CP) for high affinity/specificity. A CP-based NIR-II probe (IRT) was developed as CP-IRT for CD133+ tumor imaging with high resolution, high SBR, and specificity. CP-IRT is a peptide-conjugated small-molecule probe that can be used for targeted tumor imaging and is excreted through urine with the advantages of a low toxicity and organ uptake.

### 3.2. Tumor Vessel Imaging

In addition to targeting tumor cells, the imaging of small vessels plays a crucial role in studying and exploring the tumor microenvironment. Recently, emerging research has reported the application of advanced vascular imaging methods. Real-time vascular imaging can simultaneously obtain both anatomical and hemodynamic information, which greatly facilitates the accurate assessment of cancer detection and treatment effects [48]. The NIR-II fluorescent probe has been successfully applied to monitor tumor blood vessels in vivo, showing an extremely high penetration depth and micrometer spatial resolution.

Hao Wan et al. [49] developed a bright organ nanofluorophore (p-FE) for high-performance biological imaging in the NIR-II window. Benefiting from ultra-bright fluorescent probes and ultra-penetrating NIR-II imaging, they achieved unprecedented 3D imaging of blood vessels using single-photon NIR-II technology. When the imaging depth reaches 1.3 mm, high-contrast displays can be obtained in 5–7 μm wide micro-containers. Meanwhile, they found through the EPR effect that p-FE is a novel probe for tumors with an impressive T/NT signal ratio of about 12 and an excellent imaging agent for the vascular system with ultra-long blood circulation (a half-life time about 16 h). Moreover, Xiaolong Li et al. [50] explored the use of intense NIR-II emissive polyacrylic acid (PAA) modified NaLuF4: Gd/Nd nanorods (PAA-NRs) for the pure hexagonal phase and uniform size of the in vivo NIR-II bioimaging. Small blood vessel detection in vivo with a high spatial resolution (~105 μm) was successfully achieved. These findings demonstrate that these NIR-II fluorescent probes are promising contrast agents for tumor vessel imaging and angiogenesis diagnosis.

### 3.3. Tumor Lymphatic Imaging

The accurate detection and identification of lymph nodes by direct imaging methods could have significant therapeutic and prognostic implications for cancer diagnosis. Larger lymph nodes can be detected through the current cross-sectional imaging methods, but there may be metastases in nonenlarged lymph nodes, and not all enlarged lymph nodes are malignant. Therefore, it is still difficult to assess lymph node metastases.

Xiaoxiao Fan et al. [51] reported organic nanoprobes (IDSe-IC2F nanoparticles (NPs)) with excellent photothermal properties which could effectively label lymph nodes and help achieve high-contrast lymphatic imaging. Lingfei Lu et al. [52] presented bioluminescence probes (BPs) with emissions in the second near infrared (NIR-II) region at 1029 nm by utilizing bioluminescence resonance energy transfer (BRET) and two-step fluorescence resonance energy transfer (FRET) with a specially designed cyanine dye FD-1029. Compared to NIR-II fluorescence imaging and conventional bioluminescence imaging, the biocompatible NIR-II-BPs have been successfully applied to vessels and lymphatics imaging in mice, with 5 times higher signal-to-noise ratios and 1.5 times higher spatial resolution. Taking advantage of the Suzuki reaction, Yufeng Wang et al. [53] synthesized a novel near infrared (NIR)-II probe named TQTPA [4,4′-((6,7-bis(4-(hexyloxy) phenyl)-[1,2,5]thiadiazolo [3,4-g]quinoxaline-4,9-diyl)bis(thiophene-5,2-diyl))bis(N,N-diphenylaniline)]. They also prepared multimodal nanoparticles (NPs) loading TQTPA and cis-dichlorodiammine platinum (CDDP) (HT@CDDP) by utilizing hyaluronic acid. The NPs possessed good stability and water solubility and were sensitive to pH/hyaluronidase. Benefiting from a good tissue penetration quality and active targeting ability, the NPs can draw the outline of metastatic lymph nodes in nude mice using IR-808 under NIR exposure.

In addition, there are enormous applications for ICG in pinpointing sentinel lymph nodes (SLNs) for specific surgical resection, and several clinical trials are currently underway. ICG showed a superior tracer technique to improve the detection rate of pelvic SLN in breast cancer, cervical cancer, and gastric cancer [54,55,56]. It can reduce lymphedema to selectively remove SLNs; this can prevent tumor metastasis. ICG is clinically used to monitor the lymphatic vessels and SLNs of tumors. Alexander L Antaris et al. [40] achieved the imaging of lymphatic vasculature, lymph nodes, and lymphatic drainage tumors in U87MG tumor-bearing mice. Compared with ICG mice in the NIR-I window, the S/B ratio of the lymphatic vasculature of the CH1055-PEG-injected mice showed a clearer structure, with a nearly two-fold improvement in lymph node SBR. Interestingly, in significant contrast to the low tumor uptake of ICG, high tumor uptake was observed in CH1055-PEG-injected mice and showed a T/NT ratio as high as five at 24 h post-injection.

### 3.4. Microscopy Imaging and Optical Detection

Observing deep tissue information in vivo at a cellular or subcellular resolution to explore molecular signals and cell behaviors is a crucial direction for oncology and the study of other biological processes. In vivo imaging can provide quantitative and dynamic visualization/mapping in tumor biology and immunology. New microscopy techniques combined with NIR fluorophores provide additional options to challenge the potential space for further progress in in vivo bioimaging. Based on the development of chemical materials and physical optoelectronics, new NIR-II microscopies are emerging, such as confocal and multiphoton microscopy and light-sheet fluorescence microscopy (LSFM) [57]. Combined with suitable NIR dyes, NIR-II microscopies could benefit the observation of tumor heterogeneity.

Based on the layer scanning of confocal light, Dai Hongjie’s research group introduced elements such as near-infrared lasers (808 nm, 980 nm, 1064 nm), near-infrared photomultiplier tubes (PMT) (1000–1700 nm), and signal amplifiers into the optical path scanning unit of a traditional confocal microscope and built a new NIR-II confocal microscope for in vivo imaging. It showed good performance in the imaging of brain tissue [58], brain blood vessels [49], and ovarian tissue [59]. For example, polystyrene polyethylene glycol (PS-PEG) polymer was used to wrap fluorescent dye P-FE, which achieved about 1.3 mm penetration depth of the brain tissue at 10 μm resolution.

The NIR-II light wave has the characteristics of the low light scattering of biological tissue. Recently, Dai et al. developed NIR-II-LSFM with optical excitation and emission wavelengths up to approximately 1320 nm and 1700 nm, respectively, showing excellent penetration depth without the need for an invasive operation. The blood vessel was stained with fluorophore p-FE, and the detection targets were stained with rare earth metal ER and quantum dots PpS/CdS. A high-resolution imaging of living mouse tissue was realized that could reveal the abnormal tumor microcirculation and T cell movement trajectory and draw the 3-D molecular image of the binding process of the targeted programmed death ligand (PD-L1) and programmed cell death protein (PD-1) in living tumor tissue [60].

## *4.* NIR-II Fluorescence and Cancer Treatment

To address the inaccuracy of single imaging information, researchers have developed dual/multimodal imaging techniques able to guide cancer treatment [61]. There are significant advantages to using light to kill tumor cells, including that light is noninvasive and easy to control. Current applications of NIR-II fluorescent probes in tumor therapy includes photothermal therapy, photodynamic therapy, drug delivery, chemotherapy, and surgical navigation [62,63,64].

### 4.1. NIR-II Fluorescence Imaging-Guided Photothermal Therapy (PTT)

PTT is an optical therapy method that uses photothermal agents to convert near-infrared light into heat under the irradiation of near-infrared light; it can raise the temperature of tissues around cancer cells and kill tumor cells [65]. Combining PPT with NIR-II fluorescence imaging can provide a more precise treatment time for PPT and improve the therapeutic effects on tumors.

In 2020, Defan Yao et al. [66] developed a kind of squaraine-based NIR-II fluorescent probe for PTT guided by tumor NIR-II fluorescence imaging. In this study, they first prepared NIR-II squaraine dyes with a D-A-D structure by using squaraine as the acceptor unit and ethyl-grafted 1,8-naphthalene lactam as the donor unit. In order to realize the red-shift of the fluorescence emission of the dye to NIR-II, they then introduced malononitrile as a strong electron-withdrawing group and successfully red-shifted the fluorescence spectrum of the squaraine dye SQ1 into the NIR-II region. At the same time, a targeted nanoprobe bound to fibronectin was constructed to realize the effective treatment of the NIR-II fluorophore SQ1, showing excellent NIR-II imaging in angiography and tumor imaging (such as lung metastases in deep tissue). The SQ1 nanoprobe can significantly raise the temperature through photothermal conversion under the excitation light source and effectively ablate the tumor cells, thereby realizing NIR-II fluorescence imaging-guided PTT.

There are also a series of conjugated polymers that can be used for NIR-II fluorescence imaging-guided photothermal therapy, such as the DPP-TT NPs and P3 polymer. For example, in 2019, Lu Xiaomei et al. [67] prepared single conjugated polymer nanoparticles (DPP-TT NPs) with NIR-II PA/FL dual-modal imaging capability that could be used for tumor PTT therapy. The polymer DPP-TT, composed of a DPP donor and a fluorothiophene-[3,4-b]thiophene (TT) acceptor, was coated with DSPE-mPEG5000 to obtain DPP-TT NPs by nanoprecipitation. The DPP-TT NPs exhibited excellent stability, a high photothermal conversion efficiency of 45.4%, and negligible cytotoxicity and photothermal toxicity. Zhigang Xie et al. [68] designed a series of conjugated polymers by adjusting the absorption properties of the polymers. Among them, the P3 polymer was used for further experiments due to its sufficient absorption properties and superior solubility. It showed strong light absorption and excellent photothermal conversion in NIR-II photothermal conversion at the wavelengths of 808 and 1064 nm, with the efficiencies being 31.1% and 46.0%, respectively.

Different from the general enhanced permeability and retention effect (EPR), Zuwu Wei et al. [69] designed an active targeting nanoparticle (DPP-IID-FA) based on a DPP polymer. At 600–1115 nm, DPP-IID-FA exhibits excellent absorption capability, biocompatibility, photostability, and high photothermal conversion efficiency (49.5%). The experimental results for PTT treatment demonstrated that DPP-IID-FA showed strong cancer cell killing ability and a superior therapeutic effect in vivo. Cheng Xu et al. [70] reported a polymer multicellular nanoengager (SPNE) for synergistic second-near-infrared-window (NIR-II) photothermal immunotherapy. Based on penetrating the deep tissue of NIR-II photoirradiation, SPNE can eliminate tumors and induce immunogenic cell death, thereby triggering antitumor T cell immunity. This synergistic photothermal immunotherapeutic effect ultimately inhibits tumor growth, prevents metastasis, and procures immunological memory.

### 4.2. NIR-II Fluorescence Imaging-Guided Photodynamic Therapy (PDT)

PDT is also a commonly used phototherapy, in which the involved photosensitizers can achieve the irreversible antioxidant treatment of cancer cells or inflammation under light exposure [71,72]. At present, since most photosensitizers can only be activated by ultraviolet or visible light, in most clinical treatments, the use of PDT for tumor treatment is limited to the tumor surface. Therefore, it is necessary to urgently develop NIR-II fluorescent probes with PDT function.

In recent years, Fan Quli et al. have reported a new type of organic small molecule for NIR-II fluorescence imaging-guided PDT. In their work, they synthesized an organic small-molecule dye DPP-TT, which has strong fluorescence emission at NIR-II, enabling simultaneous NIR-II fluorescence imaging. Upon the single 808 nm laser-induced tri-modal (NIR-II fluorescence/photoacoustic/thermal) imaging-guided photothermal therapy, the study demonstrated that DPP-TT NPs showed remarkable performance in cancer therapy [67]. Chuanhui Song et al. [73] synthesized novel photosensitizer Y8 nanoparticles (Y8 NPs) via the nanoprecipitation method. Due to their long-wavelength absorption, Y8 NPs also have excellent imaging effects in the NIR-II region. In metastatic tumor-bearing murine models, Y8 NPs can effectively induce phototherapy and synergistically enhance antitumor immunity through local photodynamic and photothermal therapy, suppressing the growth of both primary and metastatic tumors without resulting in any apparent systemic toxicity. Recent articles have also reported the development of smart NIR-II nanoprobes, which could respond to the stimuli of the tumor microenvironment and inflamed sites [74,75,76]. These probes exhibit better penetration depth and better signal-to-noise ratios of the disease site in FL and PA imaging and minimize the side effects of PTT and PDT. In addition, it has been reported that nanoparticle-based PTT and PDT are able to not only ablate tumors, but also promote systemic antitumor immune activation by releasing tumor antigens and immune-stimulatory molecules [77,78,79]. Therefore, nanoprobes for PTT and PDT have great potential for use in anticancer immunotherapy.

### 4.3. NIR-II Fluorescence Imaging-Guided Drug Delivery and Chemotherapy

Chemotherapy is an important treatment option for cancer, but long-term chemotherapy leads to strong drug resistance and side effects in patients [80]. Thus, it is necessary to investigate the application of visualization and a controllable drug release system in therapy.

In recent years, the nanodelivery of drugs has received extensive attention due to the advantages of the EPR effect and the ability of nanoparticles to be enriched at tumors without adverse effects on normal tissues. Yufeng Wang et al. [53] developed a multifunctional NIR-II fluorescent probe for oral squamous cell carcinoma (OSCC), which successfully achieved the imaging-guided therapy and detection of metastatic lymph nodes. In this work, they firstly synthesized TQTPA, an organic small-molecule dye with strong NIR-II fluorescence emissions and photothermal properties that can be used as an NIR-II fluorescence imaging contrast agent and a photothermal agent. Then, TQTPA and the chemotherapeutic drug CDDP (a first-line chemotherapeutic drug for OSCC patients) were coated with hyaluronic acid (HA) by means of nanodeposition, and the TQTPA and CDDP was able to be specifically delivered to the tumor site. Based on the advantages of nanodrug delivery, Jie Li et al. [81] also designed a new type of drug-delivery nanosystem for NIR-II fluorescence imaging-guided tumor therapy. The semiconducting polymer PFTDPP and nitric oxide (NO) donor SNAP were encapsulated with an amphiphilic polymer. Under the excitation of near-infrared light, the polymer PFTDPP can generate NIR-II/PA imaging signals and realize photothermal conversion, which has the dual effects of imaging and PTT; the heat energy released at the same time becomes the stimulation source of NO release, thereby realizing synergistic therapy of the NO/PTT on lesions.

In order to reduce the degradation and inactivation of bioactive drugs (such as peptides and proteins) in the gastrointestinal tract, Rui Wang‘s group [82] reported a DCNP-based NIR-II fluorescent mesoporous microcarrier SiO_2_-Nd@SiO_2_@mSiO_2_-NH_2_. @SSPI for protein-based oral drug delivery. The carrier can not only load the drug, but also endure the harsh conditions of the intestinal tract and can, to the maximum extent, deliver the drug to the lesion. More importantly, the carrier is based on lanthanide-based down-conversion nanoparticles, which can generate a fluorescent signal at NIR-II after absorbing near-infrared light, thereby semiquantitatively monitoring drug release in vivo. Fluorescence imaging shows that the carrier can achieve a long-term stay in the intestinal site for 72 h, and the drug release amount can reach 62%, while the nontarget organs such as the liver, kidney, etc., present with low drug residual concentration. The results suggested that the toxicity and side effects are minimal, which provides novel insights into the combination of oral drugs and bioimaging. In addition, Changping Ruan et al. [83] designed a hydrogel based on a cisplatin-loaded conjugated polymer (poly(N-phenylglycine)) with NIR-II absorption ability and NIR-responsive and thermal-responsive properties. Under NIR-II laser irradiation, the localized photothermal effect not only eliminated highly metastatic triple-negative breast cancer (TNBC), but also promoted the gel-to-sol transition, triggering the on-demand release of cisplatin, which enhanced the antitumor effect and reduced drug off-target toxicity. This work not only demonstrates that the combination of PTT and chemotherapy is an effective approach to treat TNBC and improve antitumor efficiency, but also provides a promising strategy for the rational design of NIR light-responsive hydrogels for the treatment of highly aggressive cancers.

### 4.4. NIR-II Fluorescence Imaging-Guided Surgical Navigation

Cancer surgery entails surgeons having the capability to visually distinguish cancer from normal tissues under standard white-light illumination, but it is always difficult to identify precisely. Accordingly, it is of paramount importance for exploring effective approaches to accurately delineate the tumor margin. Fluorescence-guided surgery facilitates the enhancement of such capabilities by using fluorescent structure in series and color-coding the surgical field with overlaid contrasting pseudo colors from real-time intraoperative fluorescence emission. Kenneth S Hettie et al. [84] established the feasibility of using the tumor-targeting immunoconjugate (cetuximab-IRDye800) to afford improved tumor margin delineation by providing two-fold higher tumor-to-background ratios (TBRs) via utilizing the NIR-II spectral region to capture off-peak fluorescence emission from a fluorescent construct having NIR-I peak fluorescence emission.

ICG-based near-infrared imaging is becoming an important tool for many surgical procedures. The National Comprehensive Cancer Network (NCCN) recommends fluorescent SLN mapping for endometrial cancer by cervical ICG injection directly [85]. Pelvic/para-aortic sentinel lymph nodes were three times more likely to be extracted than non-SLNs, providing accurate staging for optimized postoperative patient management. Image-guided surgery using real-time ICG-fluorescence could help surgeons define tumor-negative margins in minimally invasive liver metastasectomies [86]. The optical filtering of ICG’s NIR-II photons should improve optical penetration depth, feature size resolution, and signal-to-noise ratios in these and all other ICG surgical applications. For example, ICG fusion imaging could detect occult small-sized lesions that had not been diagnosed preoperatively. In addition, ICG is effective in the real-time assessment of surgical margins by evaluating the integrity of the fluorescent rim around a colorectal liver metastases [87]. Allowing surgeons to tune intraoperatively to specific fluorescence detection bands in the NIR-I/II wavelength region should allow for visualization to be tailored to specific surgical tasks. Zhenhua Hu et al. [64] performed the first in-human liver-tumor surgery guided by multispectral fluorescence imaging in the visible and near-infrared-I/II windows and found that compared with NIR-I imaging, intraoperative NIR-II imaging provided a higher tumor-detection sensitivity, a higher tumor-to-normal-liver-tissue signal ratio, and an enhanced tumor-detection rate. Qiuxiang Wen et al. [88] constructed a novel chain-like NIR-II nanoprobe (APP-Ag_2_S-RGD), which exhibited a higher capability for cancer cell detection based on its flexible geometry, multivalent targeting, and unique NIR-II fluorescence properties. Upon intraperitoneal injection, the signal ratio of tumor to normal tissue reached a higher level, and the nonvascularized tiny tumor metastasis foci with a diameter of about 0.2 mm could be easily eliminated under the guidance of NIR-II fluorescence imaging. Therefore, the combination of the NIR-II spectral windows and suitable fluorescence probes might greatly improve image-guided surgery in the clinic.

## *5.* Conclusions and Future Directions

This review systematically summarizes the recent advances of NIR-II fluorescence probes in cancer diagnosis and treatment. NIR-II fluorescence imaging has the characteristics of low autofluorescence, high temporal and spatial resolution, and deep penetration depth, all of which have the potential to diversify bioimaging applications. Although inorganic NIR-II fluorescent probes have a high fluorescence quantum efficiency, their potential long-term biological toxicity limits their clinical translation. Organic small molecule NIR-II fluorophores have become promising candidates for clinical applications due to their high biocompatibility, excellent optical properties, rapid excretion, and easily regulated chemical structures. The inherent properties of organic small molecules give them the potential for use in photothermal therapy, photodynamic therapy, drug delivery, and surgical navigation. However, organic small molecule NIR-II fluorophores also have shortcomings, such as a low fluorescence quantum efficiency, short emission wavelength (below 1300 nm), complex chemical synthesis, and complex synthesis, all of which hinder the diversity of the bioimaging applications. Conjugated polymer-based NIR-II fluorophores have the advantage of possessing desirable optical properties, facile modification, and multifunctional carriers. They have also been shown to have efficient NIR-II fluorescence, photoacoustic and photothermal properties, and a long retention time in tumor tissue. However, the challenge remains to improve their biocompatibility, increase renal excretion, and reduce the dose for imaging and therapy. Besides this, adequate small molecule fluorophores with bright emission or activatable ability at the NIR-II window are still lacking. Therefore, the exploration of NIR-II fluorophores has reached a transformative stage, not just for basic research, but for much more.

NIR-II imaging has entered a transformative phase that is beyond basic research as well. With the development of fluorophore synthesis and a navigation system, NIR-II imaging technology will acquire rapid clinical translation. To date, all the NIR emitters in the microscopy technique field are in the preclinical phase. The optimization of the existing emitters and the exploration of the formation of a new type with both optical properties and biological properties need further exploration. Therefore, a combination of advanced NIR-II fluorescence and beam shaping based on microscope technology will prove to be a long-term goal in future. The discovery that clinical infrared dyes show bright emission tails over 1000 nm provides exciting opportunities for enhanced surgical imaging. We speculate that the NIR-II fluorescent probe will have bright application prospects in translational clinical treatment in the future, especially in clinical applications where surgeons need to assess tumor margins and make a precise resection. Integrating an InGaAs camera into an existing imaging navigation platform can provide a wide range of NIR-II spectral imaging capability, thereby greatly improving imaging quality and surgical precision. Compared with single-modality imaging, a comprehensive diagnostic system combining near-infrared imaging and other deep-penetrating imaging modalities will enable more precise detection of and efficient surgical treatment for cancer. The development of optical imaging modalities, such as photoacoustic imaging, will effectively increase the imaging depth in the NIR-II window.

## Figures and Tables

**Figure 1 biomolecules-12-01044-f001:**
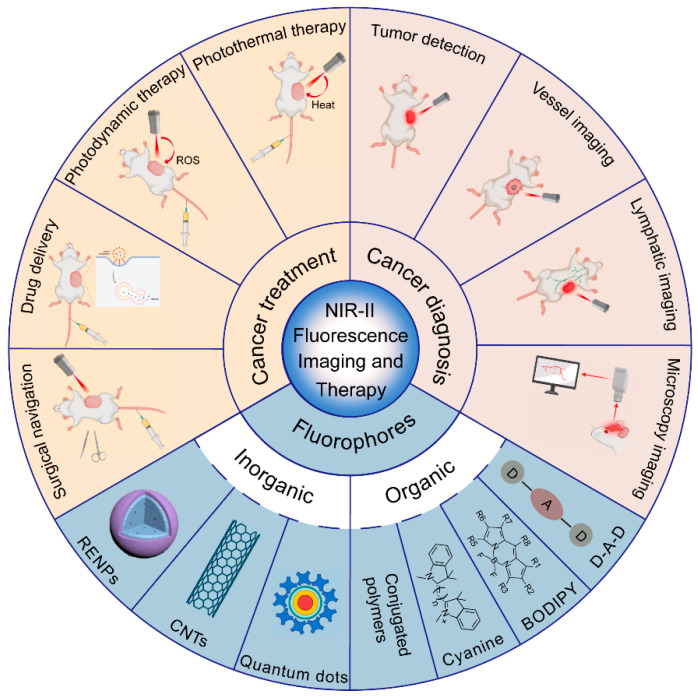
Summary of the categories of NIR-II fluorophores and applications for NIR-II fluorescence imaging in cancer diagnosis and therapy (created with BioRender.com, accessed on 27 July 2022).

**Table 1 biomolecules-12-01044-t001:** The summary of the advantages and disadvantages of NIR-II fluorophores.

NIR-II materials	Advantages	Disadvantages
Inorganic		
QDs	narrow emission wavelength, broad excitation wavelength, superior quantum yield, steady optical properties, long fluorescence lifetime	toxicity(containing heavy metal elements Cd and Pb)
CNTs	good photostability	high excitation intensity, low quantum yield
RENPs	narrow emission spectra, high emission efficiency, low photo-bleaching	toxicity, low water solubility
Organic		
organic small molecules	high biocompatibility, rapid excretion capacity, excellent resolution	low photostability low water solubility
conjugated polymers	rapid emission rate, higher fluorescence brightness, good photostability, good biocompatibility, good water dispersibility	limitation of excitation and emission wavelength

## Data Availability

Not applicable.

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
