# Peer review of "Recent Progress in Second Near-Infrared (NIR-II) Fluorescence Imaging in Cancer"

_biomolecules, 2022, doi:10.3390/biom12081044_

Round 1

Reviewer 1 Report

The subject of this review has been deeply investigated in recent literature. See for instance Frontiers in Chemistry, Research Topic: NIR-II Fluorophore and Imaging, 2021 where the corresponding author published a review entitled Recent Advances in Second Near-Infrared Region (NIR-II) Fluorophores and Biomedical Applications. I find that the current review Recent progress in second near-infrared (NIR II) fluorescence imaging in cancer submitted by the same corresponding author overlaps extensively with that of review already published. This is why I do not recommend it for publication in Biomolecules.

Reviewer 2 Report

Summary

The paper presents a representative sample overview of the recent progress in second near-infrared (NIR II) fluorescence imaging in cancer for diagnosis and therapy. This includes a nonexhaustive sample of imaging and therapeutic technologies, e.g. tumor detection, tumor microvasculature delineation, and lymphatic imaging. For therapy, it considers photothermal therapy, photodynamic therapy, chemotherapy, and surgery. It lifts and discusses fluoroprobes that are directly applied and those that are applied by nanotechnology means. The importance of the paper is that it discusses the most recent development which appears to present great promise. The downside is that it fails to clearly isolate the challenges and further research opportunities. Hence the recommendation of a section that specifically does that.

1

GENERAL COMMENTS

Reference line

1.1

This is a good paper to publish in the journal. However, a lot needs to be done before we reach that stage.

N/A

1.2

There are still too many typographical errors, e.g spacing between text and reference square brackets, e.g (Alexander L Antaris, et al. synthesized…)

N/A

1.3

There are several English language usage corrections to be made, e.g. (Then, CH1055 was conjugated to an anti epidermal growth factor receptor (EGFR) affinity, enabling tumor-targeted molecular imaging in vivo.), and (However, it is a major problem of D-A-D organic small molecule fluorescent probes in NIR-II molecular imaging that the fluorescence quantum efficiency comes to be lower because …), etc

N/A

1.4

The authors use many acronyms that should be appropriately defined at first mention or tabulated in a list of acronyms in the paper. Furthermore, the use of acronyms is not consistent in that sometimes the full names are used and at other times the acronym is used.

N/A

2

SUBSTANTIVE COMMENTS

2.1

The circular diagram is an elegant way of summarising the content of the paper. However, the central circle should read (NIR-II Fluorescence Imaging and Therapy) instead of just (NIR-II Fluorescence Imaging)

62

2.2

The authors open the discussion in section 2 by going straight into the use of fluorescence probes without giving a definition of what they are and their significance as such. We recommend that a formal definition is first given with a contextual significance and applications.

65

2.3

The problem delineation for this paper is found as you read the paper. Sometimes it is good to set a section aside for a problem discussion which explains why the need for such a review.

N/A

2.4

It is difficult to see the rows in table 1. I also think table 1 could benefit from references so that it does not appear to be a set of opinions of the authors, but a summation of literature findings.

89

2.5

The authors point out that there are enormous applications for ICG in pinpointing sentinel lymph nodes (SLNs) for specific surgical resection, while several clinical trials are currently underway. It is not clarified what clinical trials are testing and for what purpose.

201

2.6

The authors present a balanced conclusion in that it points to the way forward while outlining the limitations and challenges. While this might work for many readers, I wonder if the limitations and challenges could be isolated so as to present a clear opportunity for further research, which is one of the key value propositions of a review article.

377

Reviewer 3 Report

In the present article, authors reviewed and summarized the recent progress in second near-infrared (NIR II) fluorescence imaging in cancer. This is a potentially interesting review and helpful to understand the current stage of NIR-II imaging and to develop new strategies for future imaging applications. Some detailed comments are listed as follows.

 1. The authors have well organized the literatures and summarized the references. However, there are few similar review articles in this topic. Can authors further point out the differences with other review paper on this theme?

2. What are the key points to consider to further improve the NIR-II fluorescence imaging in future developments? May authors can offer a detailed viewpoint in the end.

3. Also authors should highlight, still what need to improve in the case of fluorophores based NIR-II fluorescence imaging from bench to bedside translation.

4. In introduction section, cite the following reference (https://pubs.rsc.org/en/content/articlelanding/2022/bm/d2bm00692h).  In that author explained the detailed molecular imaging modalities including advantages and disadvantages.

Round 2

Reviewer 1 Report

Dear Sir,

Thank you for your reply  and arguments about the novelty of your review. I partially agree with your justification. In my opinion, the degree of overlapping  your previous publication is rather high. Therefore I can not recommend your review for publication in the current form. However, in my opinion it  would find very useful for Bimolecules Journal community you extend your review by including a new section focused on NIR-II fluorescence microscopy and optical detection technology challenges, a subject rarely addressed in literature. 
